# Properties of Resorbable Conduits Based on Poly(L-Lactide) Nanofibers and Chitosan Fibers for Peripheral Nerve Regeneration

**DOI:** 10.3390/polym15153323

**Published:** 2023-08-07

**Authors:** Nurjemal A. Tagandurdyyeva, Maxim A. Trube, Igor’ O. Shemyakin, Denis N. Solomitskiy, German V. Medvedev, Elena N. Dresvyanina, Yulia A. Nashchekina, Elena M. Ivan’kova, Irina P. Dobrovol’skaya, Almaz M. Kamalov, Elena G. Sukhorukova, Olga A. Moskalyuk, Vladimir E. Yudin

**Affiliations:** 1Institute of Biomedical Systems and Biotechnology, Peter the Great St. Petersburg Polytechnic University, Polytekhnicheskaya Str., 29, Saint Petersburg 195251, Russia; spb.kamalov@gmail.com; 2Institute of Medicine, RUDN University, Miklukho-Maklaya Str., 6, Moscow 117198, Russia; maximtrube0@gmail.com; 3Scientific Research Center, Pavlov First Saint-Petersburg State Medical University, L’va Tolstogo Str., 6-8, Saint Petersburg 197022, Russia; schemiakin.i@yandex.ru (I.O.S.); solomitskiy27@mail.ru (D.N.S.); len48@inbox.ru (E.G.S.); 4Medsi Clinic, Department of Plastic Surgery, Marata Str., 6A, Saint Petersburg 191025, Russia; dr.medvedev.g@yandex.ru; 5Institute of Textile and Fashion, Saint Petersburg State University of Industrial Technologies and Design, B. Morskaya Str., 18, Saint Petersburg 191186, Russia; elenadresvyanina@gmail.com; 6Cell Technologies Center, Institute of Cytology Russian Academy of Sciences, Tikhoretsky Ave., 4, Saint Petersburg 194064, Russia; nashchekina.yu@mail.ru; 7Laboratory of Mechanics of Polymers and Composites, Institute of Macromolecular Compounds Russian Academy of Science, Bol’shoi Prospect V.O. 31, Saint Petersburg 199004, Russia; ivelen@mail.ru (E.M.I.); zair2@mail.ru (I.P.D.); yudinve@gmail.com (V.E.Y.); 8Laboratory of Polymer and Composite Materials–SmartTextiles, IRC–X-ray Coherent Optics, Immanuel Kant Baltic Federal University, A. Nevskogo Str., 14, Kaliningrad 236041, Russia

**Keywords:** chitosan fibers, chitin nanofibrils, poly(L-lactide), conduit, peripheral nerve fibers, regeneration

## Abstract

New tubular conduits have been developed for the regeneration of peripheral nerves and the repair of defects that are larger than 3 cm. The conduits consist of a combination of poly(L-lactide) nanofibers and chitosan composite fibers with chitin nanofibrils. In vitro studies were conducted to assess the biocompatibility of the conduits using human embryonic bone marrow stromal cells (FetMSCs). The studies revealed good adhesion and differentiation of the cells on the conduits just one day after cultivation. Furthermore, an in vivo study was carried out to evaluate motor-coordination disorders using the sciatic nerve functional index (SFI) assessment. The presence of chitosan monofibers and chitosan composite fibers with chitin nanofibrils in the conduit design increased the regeneration rate of the sciatic nerve, with an SFI value ranging from 76 to 83. The degree of recovery of nerve conduction was measured by the amplitude of M-response, which showed a 46% improvement. The conduit design imitates the oriented architecture of the nerve, facilitates electrical communication between the damaged nerve’s ends, and promotes the direction of nerve growth, thereby increasing the regeneration rate.

## 1. Introduction

In recent years, there has been significant progress in comprehending the pathophysiology of injuries to the peripheral nervous system and the regeneration process. Based on new information, there is a growing focus on developing microsurgical techniques to restore nerve fibers and ultimately restore sensory and/or motor functions in the body. Despite these efforts, functional recovery often falls short of pre-injury levels, leaving this problem unresolved [1]. Damaged peripheral nerves are restored by surgical intervention, namely by applying end-to-end, end-to-side, and side-to-side anastomosis. However, this is effective only when the size of the damaged nerve segment, the diastasis, is less than 3 cm. For larger lesions, suturing can hinder nerve regeneration due to mechanical stresses on the nerve tissue. Therefore, transplants from various sources are used in such cases, or the implantation of tubular channels called conduits is used instead [2,3].

The most common method of peripheral nerve repair is autotransplantation, which is considered the “gold standard” [4,5,6,7]. The source of the material is tissues or organs from one’s own body. Autografting is a framework that provides the direction of nerve fiber regeneration, Schwann cells’ viability, and neurotrophic factors’ diffusion to the defective site [8,9,10]. However, there are several limitations regarding this method, including the need for repeated operations and the challenges presented by the differences in graft and nerve tissue size. Additionally, there is a shortage of donor material, and repairing nerve tissue can be difficult due to the invasion of scar tissue caused by fibroblast migration into the diastasis zone [6,11].

While the use of allo- and xenografts can partially address these issues, their clinical application is limited due to frequent cases of immune rejection, secondary infection, and other systemic side effects [12].

Natural and synthetic polymer-based conduits have several drawbacks when it comes to nerve regeneration. Although some of them have characteristics that are better than autografts, most of them cannot restore defects with a diastasis greater than 3 cm. In addition, their surgical application is difficult, and the decomposition products can be harmful to the living organism. If an implant is obtained from a nonresorbable material, a second surgery is required to remove it [6,7,13]. Therefore, work on obtaining constructs for nerve regeneration continues, considering the results of modern research in neurophysiology, cytology, medicine, and materials science.

It is important to note that research is aimed at developing bioresorbable conduits that can stimulate axonal growth and replace defects with large diastasis. Such constructions consist of a polymeric tube with a stem or somatic cell-filled inner channel along with intercellular matrix components and a gel-like or fibrous filler [14,15,16,17,18,19,20]. Fillers create a favorable environment for the directional movement of Schwann cells, reducing their disorientation, which is essential in the nerve regeneration process [21,22]. Schwann cells play a crucial role in forming the myelin sheath of neurons, providing support, and facilitating the ionic exchange of the neuron body with the external environment, allowing for excitation transmission along the nerve fibers. The process of nerve regeneration in small diastasis involves forming a fibrin channel between the nerve fibers, permitting Schwann cell infiltration and Bungner ribbon formation, comprised of oriented columns of laminin and aligned Schwann cells. During regeneration, axons follow the bands of Bungner to the distal end, resulting in reinnervation [23,24]. However, when there are large defects in nerve regeneration, the formation of fibrin channels and bands of Bungner is difficult, so it can be difficult for nerve fibers to cross a significant gap without external support. Conduit fillers are used for this purpose.

They can be classified as biochemical or physical. Biochemical fillers include Schwann cells, nerve stem cells, mesenchymal or embryonic stem cells, growth factors, neurotrophic factors, nucleic acids, and extracellular matrix molecules such as collagen, laminin, fibronectin, etc. [25]. Thus, biochemical fillers make it possible to create a favorable environment for axon regeneration. Physical fillers, such as gels, sponges, microfilaments, microfibers, threads, and multichannel structures and their combinations are injected directly into the canal lumen to enhance directional regeneration [7,25,26,27,28,29,30]. Such fillers enhance directional regeneration by mimicking the oriented architecture of the sciatic nerve. Dividing the internal lumen of the conduit into smaller guiding tubes reduces axonal dispersion, which usually occurs in hollow canals. Fillers also affect the adhesion, proliferation, and migration of Schwann cells, nutrient transport, infiltration of blood vessels, and limit cell infiltration that prevents axon growth [31,32].

The introduction of fibrous materials into the lumen of the conduit leads to the formation of an anisotropic pore structure, the longitudinal size of which is much higher than the transverse one, increasing the surface area of the internal canal. This creates an optimal environment topology for cell adhesion and growth [25] and enhances the formation of myelinated axons that contribute to the restoration of sensory organ function [14,18,19,20,33]. Therefore, the objective of this study was to develop the design of a resorbable conduit based on poly(L-lactide) nanofibers and chitosan composite fibers containing chitin nanofibrils as a filler and to study their impact on sciatic nerve reconstruction and the restoration of its motor function.

## 2. Materials and Methods

A bioresorbable polymer, poly(L-lactide) (PLA) (Corbion PURAC, Amsterdam, The Netherlands), with a molecular mass of 20 kDa, was used to produce tubular matrices. Chloroform was used as a solvent (Ekos-1, Moscow, Russia). The polymer was dissolved at room temperature under constant stirring for 60–90 min. The polymer concentration in the solution was 14–16 wt.%. These concentrations allowed for obtaining a solution capable of electrospinning with subsequent obtaining of conduits made of nanofibers with a pore size of 100–300 nm and optimal mechanical properties: breaking strength 4.28–4.81 MPa, deformation 39.71–48.20%, and Young’s modulus 89–98 MPa [34,35].

The nanofiber tube was electroformed on a rotating cylindrical receiving electrode. The electrode had a diameter of 1.5 mm, and the rotation speed was 1500 rpm. The feed rate of the solution was 0.5–0.7 mL/h, with a voltage of 27–29 kV, and 150 mm between the feeding and receiving electrodes. After treatment in a fixed state at T = 90 °C for one hour, the obtained tube had a wall thickness of 315–350 μm.

Chitosan-based fibers were used as a conduit filler. Chitosan is the most common derivative of the natural polysaccharide chitin (chemical formulas of chitin and chitosan are shown in Figure 1). Chitin and chitosan have valuable characteristics such as biocompatibility and bioresorbability. Chitosan is a cationic polysaccharide of a basic nature which includes links of 2-amino-2-deoxy-β-D-glucopyranose residues connected by β-(1→4)-glycoside bonds and has polyelectrolyte properties. Chitosan resorption products have a neuroprotective effect, and their physical and chemical properties allow the modelling of the physiological structure of peripheral nerves [36,37,38,39,40]. Chitosan resorption products, in particular chitooligosaccharides (COS), are beneficial for promoting cell proliferation and preventing apoptosis. This is due to the ability of COS to accelerate the cell cycle and increase the proliferative activity of Schwann cells [41,42,43].

While chitosan has several advantages, obtaining conduits based on it can be challenging due to the need to achieve a structure with sufficient mechanical properties to support metabolic processes that aid in nerve tissue regeneration while also preventing infiltration by fibroblasts and other cells until complete tissue regeneration has occurred. The incorporation of fillers in the form of nanoparticles is a promising approach for improving the mechanical properties of chitosan and promoting a favorable structure for nerve regeneration. Therefore, in this research, a material based on poly(L-lactide) nanofibers was chosen as a conduit base. Chitosan fibers as well as chitosan-based composite fibers with chitin nanofibrils were used as fillers. They provided directed axonal growth as well as electrical contact between the damaged nerve ends.

Chitosan (Biolog Heppe, Landsberg, Germany) with a molecular weight of 164.2 kDa and a degree of deacetylation of 92.4% was used to obtain chitosan multifilament and monofilament fibers. Nanofibrils of chitin from SRL Mavi Sud (Aprilia, Italy) were used to obtain monofilament composite chitosan fibers. The chitosan multifilament fibers and chitosan monofilament fibers and composite chitosan monofilament fibers with chitin nanofibrils were obtained by the wet-spinning method.

It has been observed that chitosan can be transformed into fibers via the wet-spinning method by dissolving it in solvents, as its thermal decomposition temperature is below its melting point. To achieve this, a 2 wt.% acetic acid aqueous solution was used as the solvent, while the chitosan concentration was set at 4 wt.%.

To prepare the chitosan solution, the polymer powder was pre-mixed in distilled water for 30 min. This process is necessary for the chitosan to swell, resulting in faster and more uniform dissolution. Afterwards, acetic acid was added to the chitosan aqueous solution, which was then stirred for at least 1.5 h until complete dissolution occurred, yielding a homogeneous solution. The solutions were then kept at a temperature of 4 °C for 24 h to mature, after which they were filtered to remove impurities and gels and deaerated at a pressure of 0.1 atm to eliminate air bubbles. These measures were taken to prevent clogging of the holes in the spinneret. The method of obtaining these fibers is explained in [44,45]. Their structure was described in the work [46].

To obtain accurate data on the mechanical properties of the fibers, measurements of the mechanical properties (breaking stress, Young’s modulus, and strain at break) of the fibers were carried out using an Instron 5943 tensile testing machine. The fiber length used as the base was 100 mm, and the fiber loading rate was maintained at 10 mm/min. Prior to testing, it was ensured that the threads had been stored in normal climatic conditions for a minimum of 24 h, with a relative humidity of 66% and a temperature of 20 ± 2 °C.

The swelling of the fibers was assessed by determining their diameter using a ZEISS Axio Scope.A1 optical microscope (Hilden, Germany) at 10× magnification, following a period of immersion in PBS for 3, 7, 14, 30, 60, and 90 days.

For the in vivo experiments on rat sciatic nerve regeneration, PLA nanofiber tubes were used as conduits, as well as tubes containing chitosan multifilament fibers, monofilament fibers, and composite monofilament fibers containing 5–50 wt.% chitin nanofibrils. The diameter of the monofilament fibers ranged from 45–50 μm, while that of the filaments in multifilament fibers was 20–25 μm. The number of fibers introduced into the conduit was such that it occupied 50% of the free volume in the conduit’s inner channel. The free volume is necessary for the directed growth of Schwann cells and the transport of proteins required for axon regeneration. Based on the geometrical dimensions of the conduit and filler, the monofilament fibers or multi-filament fibers were inserted into tubes. Figure 2 shows the model, and in Figure 3 are photos of the conduit filled with chitosan monofilament fibers (made by scanning electron microscope (SEM)).

Human embryonic bone marrow mesenchymal stromal cells (FetMSCs) cell lines were used to test the biocompatibility of the PLA matrixes in vitro (5 passages). The cells were cultured in a CO_2_ incubator at 37 °C in a humidified atmosphere containing air and 5% CO_2_ in DMEM/F12 nutrient medium (Gibco, Waltham, MA, USA), 10% (*v*/*v*) thermal inactivated fetal bovine serum (FBS; HyClone, Logan, UT, USA), 1% L-glutamine, 50 U/mL penicillin, and 50 µg/mL streptomycin. For the experiment, a 200 μL cell suspension containing 20,000 cells was applied to the samples, incubated for 2 h, and then filled with 5 mL of complete nutrient medium and cultured for 1 day in a CO_2_ incubator at 37 °C in a humidified atmosphere containing air and 5% CO_2_. After the indicated time, the medium was removed, and the cells were washed with phosphate buffer solution and fixed with 4% formalin solution, followed by treatment with ethanol solution.

In vivo experiments were performed on male white Wistar rats weighing between 180 and 200 g, aged 3 months, and there were 10 animals in each group. The animals underwent surgery under general anesthesia (Zoletil (Virbac, Carros, France) 100–0.1 mL and Rometarum (Bioveta Inc., Ivanovice na Hané, Czech Republic) 20 mg/mL solutions—0.0125 mL per 0.1 kg of animal weight, intraperitoneally). All experiments followed the principles of the European Convention for the Protection of Vertebrate Animals used for Experimental and Other Scientific Purposes (Strasbourg, 1986) and the Declaration of Helsinki of the World Medical Association (Helsinki, 1996).

Rats were randomly divided into the following seven groups (n = 10 in each group): nerve autografts (AG), PLA conduits without fibers (PLA), PLA conduits filled with multifilament chitosan fibers (PChs), monofilament chitosan fibers (MChs), and monofilament chitosan fibers filled with chitin nanofibrils at 0.5 (Chs-ChNF 0.5%), 30 (Chs-ChNF 30%), and 50 (Chs-ChNF 50%) wt.%.

The procedure for implanting the conduits into the rat sciatic nerve is illustrated in Figure 4. Under general anesthesia, the sciatic nerve was injured, and a 10 mm gap was created at the level of the middle of the femur. In the control group (AG), the resected distal segment of the nerve was sutured to the proximal end of the nerve and vice versa. In the other groups, the diastasis was replaced with conduits. After the implantation, the wounds were sutured in layers with Prolen 4-0 thread and atraumatic needles. After the surgical operation, the animals were kept in special conditions for four months with free access to water and food and the ability to move freely. All animals were active, and no negative effects of the implanted materials were observed, as evidenced by their general health and the absence of inflammatory processes in the implanted area.

### Assessment of the Degree of Nerve Conduction Recovery

Electroneuromyography (ENMG) is widely regarded as the most objective and reliable method for evaluating the functional state of peripheral nerves [47]. ENMG involves the recording and analysis of bioelectric potentials in both muscles and peripheral nerves. By exposing the individual to low-intensity electrical impulses and recording the corresponding muscle response, the ENMG procedure can evaluate the total action potential of all motor units—known as the M-response. Through this process, the amplitude of the M-response can indicate the number of motor units present in a given muscle, and a decrease in amplitude may suggest a reduction in the number of motor units. To perform ENMG, the animals were carefully immobilized on a manipulation table before reference and stimulating electrodes were inserted into the m.biceps femoris (biceps femoris), with the grounding electrode fixed on the foot.

Assessing motor function recovery after the regeneration of damaged sciatic nerves is a critical process, and this was done by determining the functional index (SFI). For this purpose, hind paw prints were fixed on pre-prepared paper strips before lowering the animals to move them along a predetermined trajectory. The parameters measured in both normal (N) and experimental (E) paws were the length of the print (PL), the length from the tip of the longest toe to the heel (TS), and the intermediate toe spread (IT), as shown in Figure 5. SFI was then calculated using the following formula:(1)$$SFI=-38.3\frac{{PL}_{E}-{PL}_{N}}{{PL}_{N}}+109.5\frac{{TS}_{E}-{TS}_{N}}{{TS}_{N}}+13.3\frac{{IT}_{E}-{IT}_{N}}{{IT}_{N}}-8.8$$

Based on the formula, it has been observed that the SFI value is registered at zero in the undamaged limb, whereas it is noted as 100 in animals that have impaired nerve function.

Motor-coordination disorders were studied using a complex “Rotarod” manufactured by Orchid Scientifics (Nashik, Maharashtra, India). The technique is based on the ability of small laboratory animals to stay on a rotating drum [48]. An experiment was conducted wherein an animal was placed in a compartment of the apparatus and the rotation speed of the drum was varied in real-time, ranging from 8 to 12 rpm. The motivating factor was the height. The score of the activity was based on the duration of time the animals stayed on the drum until they fell. The findings of the study revealed the length of time the animals were able to remain on the drum before falling, the speed of the drum’s rotation at the time of the animal’s fall and landing, and the nature of the motivating factor.

To study the recovery of neuromuscular functions, an original technique based on measuring the force of grasping an object by the limbs using a force meter from Orchid Scientifics (Nashik, Maharashtra, India) was used [49]. The device is equipped with a digital force sensor that accurately measures the desired parameter. The grip force sensor is connected to a wire mesh that is affixed to an anodized base plate. The standard sensor has a capacity of up to 19.6 H. The animals were carefully guided towards the grid until they were able to grip the wire with their test limbs. The animals were carefully guided towards the grid until they were able to grip the wire with their test limbs. While holding their torso parallel to the mesh, the animals were gently drawn backwards, away from the mesh. The pace was kept at a level that allowed for the development of resistance to the pulling force. The peak force values were recorded at the precise moment the mesh was released.

## 3. Results

### 3.1. Physical and Mechanical Characteristics of Chitosan Fibers

Mechanical properties of the multifilament and monofilament fibers are presented in Table 1.

It has been observed that composite fibers possess notably higher strength and modulus of elasticity when compared to pure chitosan fibers. It has also been noted that fibers containing 0.5 wt.% of chitin nanofibrils exhibit the best mechanical characteristics. This could be attributed to the orientation of chitosan macromolecules along chitin nanofibrils during fiber spinning, as mentioned in [1,2]. Additionally, it is important to note that an increase in the concentration of chitin nanofibrils results in a decline in the mechanical properties of composite fibers. This can be explained by the formation of a rigid network of filler particles, resulting in preventing the orientation of chitosan macromolecules during fiber drawing.

The results of the investigation into the swelling of multifilament and monofilament fibers with varying chitin nanofibrils content in PBS can be observed in Table 2.

Over the course of 30 days in PBS, there was a gradual increase in the diameter of the fibers. It was observed that on days 60 and 90, there was a slight increase in diameter. When the ChNFs content in the fiber increased, there was a decrease in the swelling in PBS. This is because ChNFs are resorbed more slowly compared to chitosan. It should be noted that fibers without ChNFs or with a ChNFs content of 0.5% on the 30th day begin to lose their shape. Additionally, on the 90th day, there was an increase in the spread of values.

### 3.2. Biocompatibility of Conduits

The biocompatibility of PLA conduits filled with chitosan fibers was assessed in this study. PLA has been previously proven to be an advantageous material for tissue engineering structures, including nanofibers, as it has demonstrated no cytotoxicity with various cell assessment methods in prior studies [50].

In an in vitro study [51], PLA nanofiber tubes were found to have significantly more surface area than smooth-surface tubes, affecting protein adsorption and cell adhesion. The results of this study also showed enhanced adhesion of rat PC12 cells and rabbit patellar fibroblasts to the nanofiber scaffolds compared to smooth-surface scaffolds. Several works have also demonstrated the impact of the complex PLA-based surface topology on cell adhesion and proliferation, as well as differentiation and phenotype changes [52]. The biocompatibility and lack of cytotoxic effects of the PLA material utilized in this study have been demonstrated in the literature [53]. Therefore, the focus of this study was solely on evaluating the effectiveness of cell–cell interaction with the nanofiber-based surface.

Human mesenchymal stromal cells were used to assess biocompatibility. The photos in Figure 6 show that the cells completely covered the end part of the conduit and occupied nearly the entire surface within the first day after seeding.

The extensive area of the melted cells indicates a high affinity of living cells for this material, and it is reasonable to assume that native tissue cells will migrate to this surface in the damaged area and lead to a relatively rapid integration of the formed construct with the patient’s body after transplantation. Based on the results obtained, it can be concluded that the filled conduits have no toxic effect on FetMSCs.

### 3.3. Nerve Conduction Recovery Assessment

The results of the ENMG study of the animals in which the sciatic nerve was reconstructed using PLA nanofiber conduits are presented in Table 3.

Based on the data presented in Table 1, it appears that the amplitude of the M-response decreased significantly in all animals of the experimental groups after 4 weeks of observation. This suggests a notable decline in neuromuscular conduction in the hind limbs compared to the intact limbs. However, after 16 weeks, there was an increase in M-response amplitudes in all cases except for the conduit with PChs. Higher M-response amplitudes were observed in groups PLA/MChs and PLA/Chs-ChNF 50%. These results are comparable to those found in group AG (particularly when the “gold standard” method of autografts was used). It has been observed that the existence of chitosan monofibers has a favorable impact on the speed of guided axonal growth, regeneration rate, and nerve myelination. This can be attributed to two factors: firstly, the creation of inner fiber space in the form of longitudinal channels and the structuring of free tube volume. Chitosan monofibers possess a high elastic modulus, up to 7.8 GPa, and thus lend stiffness to the structure formed inside the tube. The proper arrangement of the filler structure in the longitudinal direction, ideal pore size, and their alignment along the tube axis contribute to the directed growth of Schwann cells and the formation of the nerve structure in proximity to the native one. Secondly, a significant feature of chitosan and chitosan-based fibers is their electrolyte properties [54]. The existence of NH3+ ions in the liquid medium provides ionic conductivity, and hence promotes communication between the nerve ends. Consequently, the existence of chitosan-based fibers in the conduit facilitates the regeneration of the sciatic nerve with 10 mm diastasis.

### 3.4. Motor Recovery Assessment

Assessment of motor function recovery after the regeneration of the injured sciatic nerve was conducted by means of determining SFI at 16 weeks following implantation (Figure 7). In an unimpaired nerve, the SFI is zero, whereas in a non-functioning nerve, the SFI is approximately 100.

It should be noted that in all cases of nanofiber conduits, the SFI value is slightly higher than that of autonervous conduits. However, the conclusion on the merits of the conduits for sciatic nerve repair can be made after long-term observation of the animals, using stimulating electrical and mechanical influences.

The results depicted in Figure 7 illustrate that the SFI is less than 100 at 16 weeks after implantation of the conduits, regardless of whether they contain fiber fillers or not. This suggests that the process of sciatic nerve repair is taking place. Notably, the most significant progress in nerve repair was observed in the groups PLA/MChs and PLA/Chs-ChNF 50%. Conversely, the process of nerve repair was slower in conduits without fiber filler or with multifilament filler (PLA/PChs). This highlights the positive impact of the conduit’s inner structure, especially the channels formed in the inter-fiber space, which facilitate the directed growth of regenerating axons, thereby influencing their speed and number.

It is important to mention that in all instances of monofilament fiber conduits, the SFI is slightly higher than that of autografts. Nevertheless, further long-term observations of animals using stimulating electrical and mechanical influences are required to conclude the advantages of conduits for sciatic nerve repair.

### 3.5. Investigation of Motor-Coordination Disorders

The evaluation of motor coordination activity was determined through the rotation speed of the drum. This was measured by the speed at which the animal could no longer maintain its position on the drum. The initial rotation speed was set at 8 rpm, with a gradual increase of 1 rpm every 5 s. The diameter of the drum was 6 cm, and the motivating factor for remaining on the drum was the height, which was 30 cm. The findings of motor-coordination disorders are depicted in Figure 8 and Figure 9.

It has been observed that rats with autografts were able to remain on the rotating drum at a speed of up to 12 rpm, which is similar to the value for the intact nerve (as seen in Figure 9). Additionally, in groups where the conduit design included chitosan-based monofibers, the animals were able to stay on the surface of the rotating drum up to a rotational speed value of 9–11 rpm. It was also noted that as the content of chitin nanofibrils in the composite fiber increased, the value of limiting velocity increased as well. However, the absence of fibers in the conduit resulted in a decrease in the value of angular velocity.

The presence of chitosan fibers with chitin nanofibrils accelerates motor coordination recovery. This is due to the better regeneration of nerve fibers and the preservation of more axons. In the group with polyfilament filaments, the animals failed to fixate on the drum rotating at a minimum speed of 8 rpm. This suggests the ineffectiveness of using a filler in the form of chitosan polyfilament filaments.

In the group PLA/PChs, positive outcomes were not observed. The animals in this group failed to fixate on a rotating drum at a minimum speed of 8 rpm, indicating that this approach may not be effective.

Similarly, the use of hollow conduits resulted in the sluggish movement of the animals at low rotational speeds and the inability to stay on the drum at speeds greater than 8 rpm. Although the recovery of motor coordination functions was observed in this group, it was not of the same quality as that of animals for which conduits with chitosan monofilaments were used.

### 3.6. Neuromuscular Function Recovery Assessment

The extent of recovery was determined by comparing the gripping force of the injured paw with that of the undamaged one. The findings of the study are presented in Table 4.

As per Table 4, the animals with autografts exhibited the highest gripping force. For the conduits, the most promising results were observed in groups PLA/MChs and PLA/Chs-ChNF 50%. The degree of recovery was calculated by comparing the grip strength of the limb with the implant to that of a healthy limb, which suggests that the conduits with chitosan monofilament fibers were effective.

It has been observed that both autografts and implantation of tubular conduits did not lead to complete recovery. However, the introduction of fillers has shown promising results similar to the autoneural insertion method. It is worth considering that the restoration rate of the injured limb’s functions has an impact on the motor activity of the intact part. Thus, if the injured leg functions poorly, the state of the intact one worsens due to a decrease in the mobility of the animal. For instance, in the groups where conduits with monofilament fibers and autografts were used, the strength of gripping an object by the intact limb was 10–13% higher than the same parameters in the PLA (hollow) group.

## 4. Conclusions

Poly(L-lactide) nanofiber conduits with chitosan fibers or composite chitosan fibers (containing 0.5–50.0 wt.% chitin nanofibrils) on the inner surface have been developed as resorbable conduits for peripheral nerve regeneration. It was shown that these nanofiber conduits with monofilament chitosan fibers serve as guiding elements for axon growth.

It was shown that monofibers based on chitosan and chitin nanofibrils with elastic modulus E = 7.8 GPa promoted directed neuron growth. In addition, it has been suggested that the ionic conductivity of chitosan contributes to the communication of neurons of the peripheral nerve ends.

Furthermore, it has been observed that the use of chitosan monofibers increases the amplitude of the M-response, as determined by studies of nerve conduction recovery.

Importantly, the safety of these conduits was demonstrated by the absence of toxic effects on FetMSCs.

The findings of the study on motor coordination activity have indicated an accelerated recovery of damaged nerves in the presence of chitosan monofilaments within the conduit.

The assessment of SFI has confirmed the presence of functional recovery of the sciatic nerve when the conduit is filled with monofibers, with the SFI ranging from 76 to 83.

Moreover, the study on neuromuscular function recovery has shown a positive impact on the presence of chitosan monofibers in the conduits. As the content of chitin nanofibrils in the chitosan monofilament fibers increases, the degree of recovery also increases to 46%, which is comparable to the results obtained using autoneural insertion for defect replacement.

Overall, it can be concluded that chitosan monofilament fibers in the tubular conduit structure have a favorable effect on the rate and degree of regeneration of damaged peripheral nerves. The content of chitin nanofibrils in chitosan composite fiber can significantly improve the hind limb functions of animals.

## Figures and Tables

**Figure 1 polymers-15-03323-f001:**
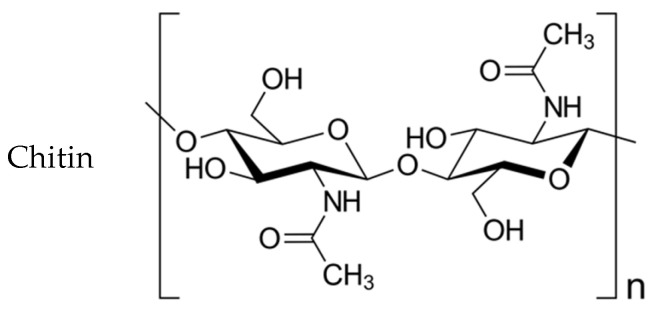
The structure of chitin and chitosan macromolecules.

**Figure 2 polymers-15-03323-f002:**
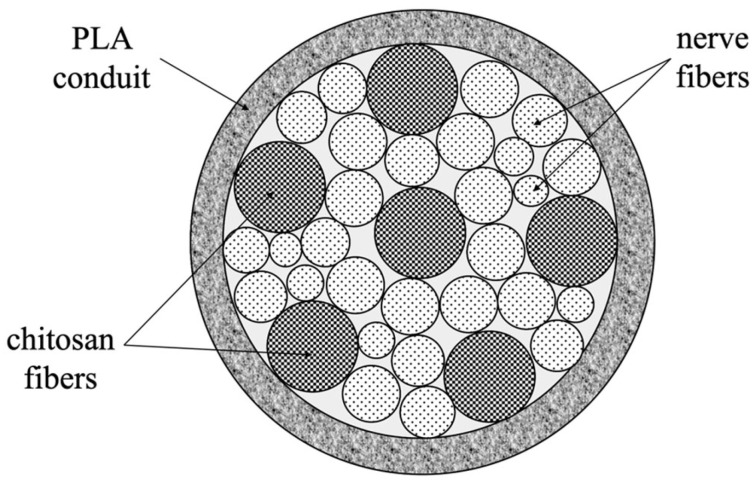
The scheme of the conduit filled with chitosan fibers.

**Figure 3 polymers-15-03323-f003:**
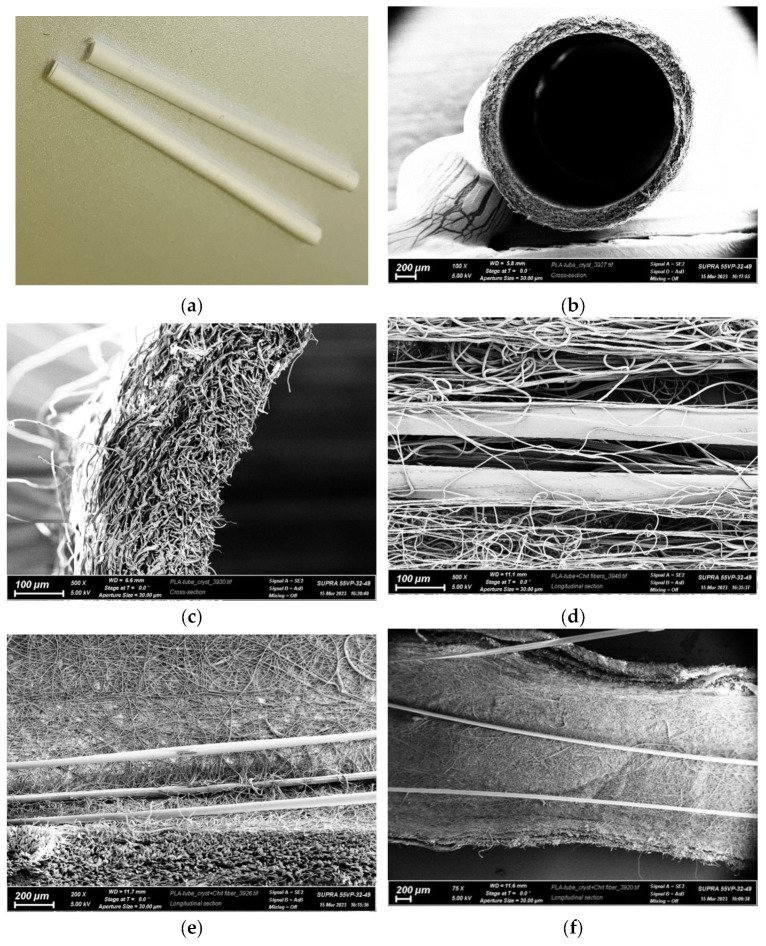
Microphotographs of a tubular conduit based on poly(L-lactide) nanofibers and chitosan monofibers; (**a**) external view; (**b**,**c**) end face of conduit based on poly(L-lactide); (**d**) internal structure of conduit with chitosan fibers before crystallization; (**e**,**f**) internal structure of conduit with chitosan fibers after crystallization.

**Figure 4 polymers-15-03323-f004:**
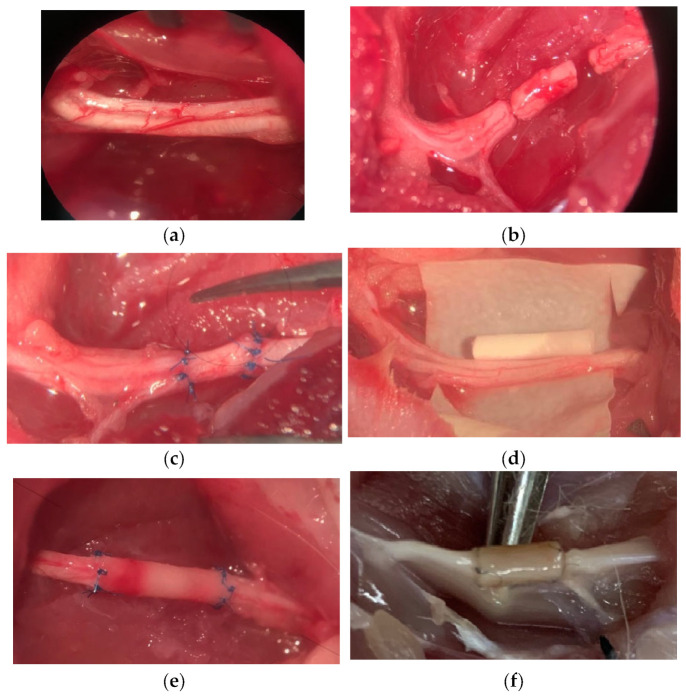
The scheme of conduit implantation into the rat sciatic nerve: (**a**) native sciatic nerve, (**b**) sciatic nerve with 10 mm diastasis; (**c**) nerve defect replaced with an autoneural insert; (**d**) PLA nanofiber-based tubular conduit; (**e**) conduit implanted in the sciatic; (**f**) conduit 4 months after implantation.

**Figure 5 polymers-15-03323-f005:**
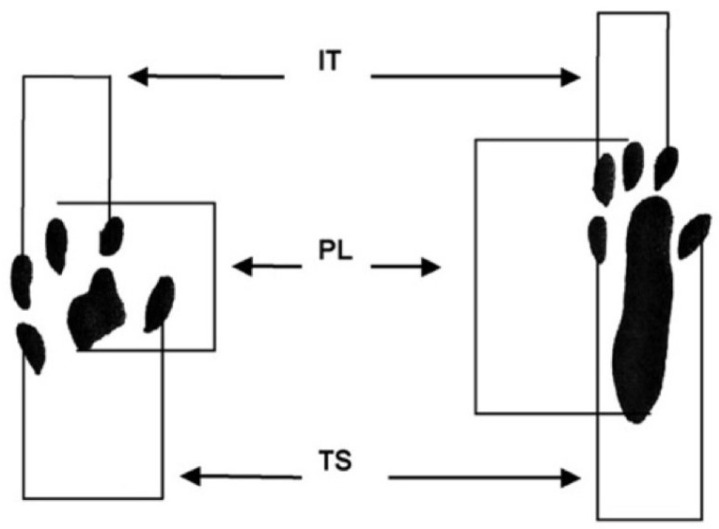
Schematic diagram of SFI measurement parameters.

**Figure 6 polymers-15-03323-f006:**
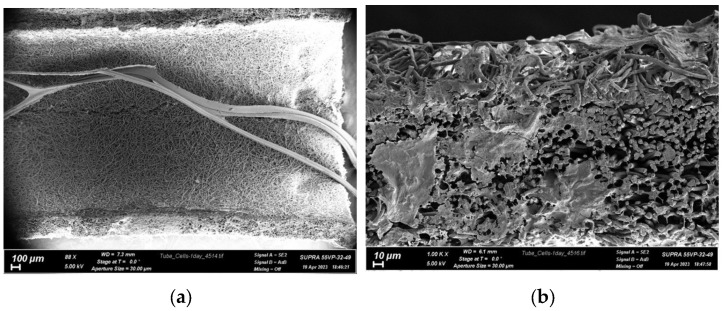
SEM photos of FetMSCs cell morphology after 24 h of cultivation on polylactide conduits filled with chitosan fibers: (**a**) general view of the inner surface of the conduit with chitosan fibers; (**b**) end face of the polylactide conduit.

**Figure 7 polymers-15-03323-f007:**
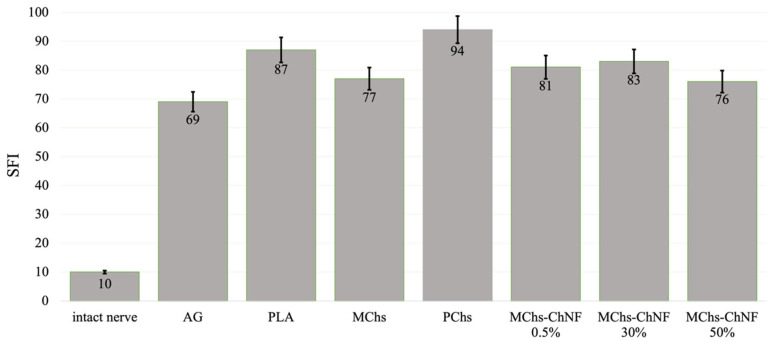
SFI at 16 weeks after implantation.

**Figure 8 polymers-15-03323-f008:**
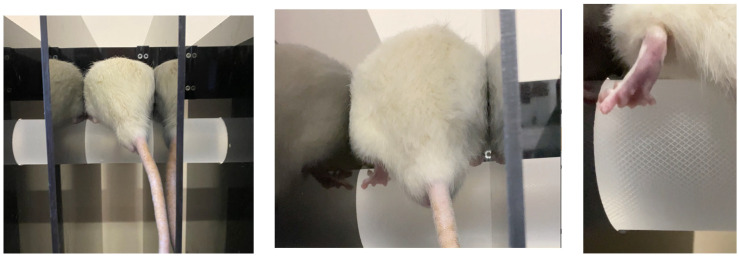
Studies of motor-coordination disorders of the animals after conduit implantation.

**Figure 9 polymers-15-03323-f009:**
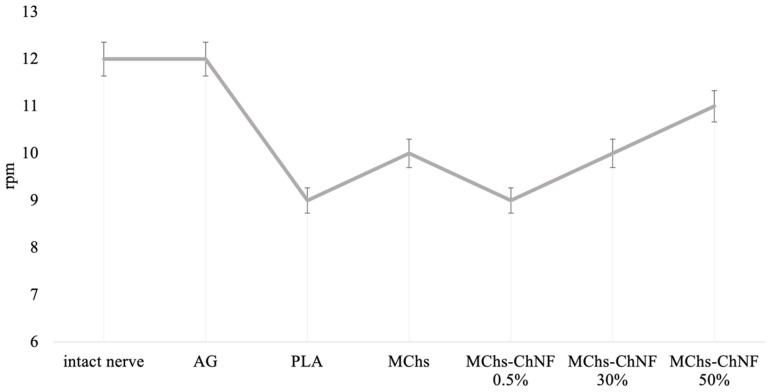
Motor activity recovery of animals.

**Table 1 polymers-15-03323-t001:** Mechanical properties of the chitosan fibers.

Fibers	Tensile Strength, σ, MPa	Young Modulus, GPa	Elongation at Break, ε, %
MChs	240.96 ± 7.98	13.83 ± 0.54	6.3 ± 0.72
PChs	186.58 ± 19.91	12.09 ± 1.9	3.36 ± 0.32
PLA/Chs-ChNF 0.5%	259.31 ± 10.45	14.41 ± 0.73	5.35 ± 0.34
PLA/Chs-ChNF 30%	174.37 ± 7.31	14.01 ± 0.81	3.87 ± 0.43
PLA/Chs-ChNF 50%	163.86 ± 9.26	13.87 ± 0.19	2.57 ± 0.55

**Table 2 polymers-15-03323-t002:** Swelling capacity of different chitosan fibers.

Fibers	Diameter, µm						
	Dry	3 Days	7 Days	14 Days	30 Days	60 Days	90 Days
PChs	24.17 ± 4.19	35.52 ± 3.91	37.55 ± 2.90	42.16 ± 3.99	65.37 ± 8.15	68.92 ± 5.21	70.34 ± 8.13
MChs	43.67 ± 3.79	52.67 ± 5.59	60.38 ± 5.21	78.23 ± 7.89	101.37 ± 7.73	102.72 ± 7.01	103.44 ± 6.93
Chs-ChNF 0.5%	42.75 ± 2.24	47.16 ± 3.67	59.74 ± 4.02	69.10 ± 3.72	95.48 ± 2.31	96.63 ± 6.29	98.04 ± 5.90
Chs-ChNF 30%	40.88 ± 1.98	60.01 ± 4.97	70.16 ± 3.18	83.87 ± 4.71	91.92 ± 5.01	93.51 ± 4.61	93.34 ± 4.01
Chs-ChNF 50%	43.10 ± 2.03	55.82 ± 3.72	75.20 ± 2.94	80.42 ± 4.71	90.15 ± 2.59	89.89 ± 4.75	91.56 ± 3.09

**Table 3 polymers-15-03323-t003:** M-response amplitude at 4 and 16 weeks after the implantation.

Conduit	M-Response Amplitude (mV)		
	Intact Nerve	4 Weeks	16 Weeks
AG	28.9 ± 3.3	6.8 ± 2.4	15.1 ± 5.2
PLA	27.3 ± 3.8	5.1 ± 1.8	8.8 ± 3.9
PLA/MChs	27.2 ± 3.6	5.5 ± 2.2	13.8 ± 3.2
PLA/PChs	27.8 ± 2.1	3.9 ± 1.5	6.9 ± 4.7
PLA/Chs-ChNF 0.5%	30.1 ± 3.1	4.3 ± 1.7	12.5 ± 5.8
PLA/Chs-ChNF 30%	28.8 ± 3.7	4.9 ± 1.4	12.7 ± 3.7
PLA/Chs-ChNF 50%	28.2 ± 2.5	5.3 ± 0.9	14.1 ± 4.6

**Table 4 polymers-15-03323-t004:** Neuromuscular function recovery.

Type if Conduit	Grip Force, N		
	Intact Nerve	Conduit	Recovery Degree, %
AG	2.77 ± 0.41 (14%)	1.54 ± 0.21 (13%)	55
PLA	1.80 ± 0.17 (10%)	0.59 ± 0.1 (17%)	33
MChs	2.54 ± 0.31 (12%)	1.14 ± 0.41 (36%)	45
PChs	-	-	-
Chs-ChNF 0.5%	1.95 ± 0.49 (25%)	0.82 ± 0.17 (21%)	42
Chs-ChNF 30%	2.06 ± 0.31 (15%)	0.96 ± 0.25 (26%)	46
Chs-ChNF 50%	2.73 ± 0.51 (19%)	1.20 ± 0.48 (40%)	44

## Data Availability

Not applicable.

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
