# Peer review of "Properties of Resorbable Conduits Based on Poly(L-Lactide) Nanofibers and Chitosan Fibers for Peripheral Nerve Regeneration"

_polymers, 2023, doi:10.3390/polym15153323_

Round 1
Reviewer 1 Report
Dear authors,
I read your paper entitled "Properties of resorbable conduits based on poly(L-lactide) nanofibers and chitosan fibers for peripheral nerve regeneration" sent to Polymers journal and here are my comments:
1. Which is the added value of the paper with respect to existent literature?
2. Was the stirring done for 60 or 90 minutes in the case of PLA dissolution in chloroform? Why did you choose 16 wt.% concentrations?
3. The paper lacks physico-chemical characterization of the specimes as it is a polymer journal.
4. It is not clear how the conduit was obtained from PLA, chitosan and chitin. Please better describe this procedure.
5. What about the swelling ability of these products in PBS or similar solutions?
6. Please add the mechanical properties of the specimens.
7. You must add the approval for in vivo experiments in your institution.
8. You must also add the approval for working with Human embryonic bone marrow mesenchymal stromal cells (FetMSCs) cell lines.
9. What about the biodegradability test?
10. Why do you thes ematerials composites? "...composite chitosan fibers (containing 0.5-50.0 wt% chitin nanofibrils)..."
11. The animal testing is a nice tool in the paper, but you have also to keep in mind to add materials' characterization and other recommended tests.
Author Response
Dear Sir/Madam,
We have received your review comments and carefully familiarized ourselves with them. We appreciate you reading our paper and providing us with helpful comments. Your comments will help us improve our paper.
Below are our responses to each of your comments:
- Which is the added value of the paper with respect to existent literature?
Tubular bioresorbable conduits have been developed for the regeneration of peripheral nerves. The conduits consist of poly(L-lactide) nanofibers and chitosan composite fibers with chitin nanofibrils, which were obtained and added to the inner structure of the tubular conduit.
It is hard to find studies on the effect of chitin nanofibrils and chitosan on nerve regeneration, although it is known that chitosan is a cationic polysaccharide and chitooligosaccharides (resorption products) which have a neuroprotective effect, accelerate the cell cycle, and increase the proliferative activity of Schwann cells.
Moreover, chitosan fibers with chitin nanofibrils are oriented in the internal structure of the conduit, which adds the ability to direct the growth of axons, thereby accelerating the regeneration process. So, the conduit design imitates the originally oriented architecture of the nerve, facilitates electrical communication between the damaged nerve's ends, and promotes the direction of nerve growth, thereby increasing the regeneration rate. But also, chitin nanofibrils improve the mechanical qualities of chitosan fibers.
- Was the stirring done for 60 or 90 minutes in the case of PLA dissolution in chloroform? Why did you choose 16 wt.% concentrations?
We have added that the polylactide concentration was 14-16 wt.% (lines 110-111). The polymer was dissolved at room temperature under constant stirring for 60-90 minutes depending on polymer concentration in solution, rotation speed and environmental conditions.
The polylactide concentration has been determined in previous studies as 14-16 wt.%. These concentrations allowed to obtain a solution capable for electrospinning with subsequent obtaining of conduits made of nanofibers with pore size 100-300 nm and optimal mechanical properties: breaking strength 4,28-4,81 MPa, deformation 39,71-48,20 % and Young's modulus 89-98 MPa (lines 110-114, references 35 and 36).
- The paper lacks physico-chemical characterization of the specimens as it is a polymer journal.
In this paper constructions made of several components were used for maintaining peripheral nerve regeneration. The constructions consist of a polylactide tube made of micro- and nanofibers; multi-filament and monofilament chitosan fibers, and monofilament composite chitosan fibers with chitin nanofibrils. Chitosan fibers were injected into the lumen of the polylactic conduit directly before surgery. Each of the components of the construction was studied by the author and co-authors separately earlier. The results of studies of polylactide conduits (physico-mechanical, physico-chemical properties, spinning parameters) are presented in papers 35 and 36. Studies of chitosan fibers both with and without chitin nanofibrils (determination of the optimal spinning parameters, the effect of chitin nanofibrils on the fibers’ mechanical properties) are presented in papers 45, 46, and 47. Since the results were published earlier, the article provides links to them.
- It is not clear how the conduit was obtained from PLA, chitosan and chitin. Please better describe this procedure.
The description of PLA conduit obtaining is presented in lines 107-119.
The description of the chitosan and composite chitosan fibers obtained were added at lines 142-160.
The description of the construction obtained made of PLA conduits and fibers were added at lines 170-180.
- What about the swelling ability of these products in PBS or similar solutions?
The swelling of the fibers was assessed by determining their diameter using a ZEISS Axio Scope.A1 optical microscope at 10X magnification, following a period of immersion in PBS for 3, 7, 14, 30, 60 and 90 days (lines 167-169). The results can be observed in Table 2 (line 285).
- Please add the mechanical properties of the specimens.
Measurements of the mechanical properties (breaking stress, Young's modulus, and strain at break) of the fibers were carried out using an Instron 5943 tensile testing machine. The fiber length used as the base was 100 mm, and the fiber loading rate was maintained at 10 mm/min. Before testing, it was ensured that the fibres had been stored in normal climatic conditions for a minimum of 24 hours, with a relative humidity of 66% and a temperature of 20 ± 2°C (lines 161-166). The results are observed in Table 1 (line 272).
- You must add the approval for in vivo experiments in your institution.
All experiments followed the principles of the European Convention for the Protection of Vertebrate Animals used for Experimental and Other Scientific Purposes (Strasbourg, 1986) and the Declaration of Helsinki of the World Medical Association (Helsinki, 1996).
Approval for in vivo experiments in the institution â„– 78-11573 (given to Pavlov First Saint-Petersburg State Medical University, L’va Tolstogo str. 6-8, Saint-Petersburg, 197022, Russia).
- You must also add the approval for working with Human embryonic bone marrow mesenchymal stromal cells (FetMSCs) cell lines.
The study was conducted according to the guidelines of the Declaration of Helsinki and approved by the Animal Welfare Assurance of INC RAS (IN F18-00380, 2017–2022, 9 July 2018).
- What about the biodegradability test?
We have conducted extensive degradation analysis of chitosan composite fibers with chitin nanofibrils and polylactide conduits in PBS and PBS with lysozyme. We have obtained interesting voluminous results and devoted a separate article. Unfortunately, the article is currently under review in the journal, so I am not allowed to include degradation data.
- Why do you thes ematerials composites? "...composite chitosan fibers (containing 0.5-50.0 wt% chitin nanofibrils)..."
Chitin and chitosan have valuable characteristics such as biocompatibility and bioresorbability. Chitosan is a cationic polysaccharide of basic nature. Chitosan resorption products have a neuroprotective effect, and its physical and chemical properties allow the modelling of the physiological structure of peripheral nerves. Chitosan resorption products, in particular chitooligosaccharides (COS), are beneficial for promoting cell proliferation and preventing apoptosis. This is due to the ability of COS to accelerate the cell cycle and increase the proliferative activity of Schwann cells. Although chitosan has numerous advantages, obtaining conduits based on it is limited because of the difficulties that arise in creating the aligned structure in the wet environment.
Chitosan has not got enough mechanical properties in a wet environment. To improve his mechanical properties and features the chitin nanofibrils were used to obtain composite fibers. Composite fibers still have got advantages of chitin and chitosan, moreover, they have better mechanical properties (lines 132-138).
In the previous studies coauthors determined the maximum concentration of chitin nanofibrils in the chitosan fibers as 50.0 wt%. This concentration is limited by the viscosity properties of wet-spinning solutions. To obtain fibers with a higher concentration of chitin nanofibrils was unable. Adding chitin nanofibrils less than 0.5 wt% does not improve any properties of chitosan fibers.
- Animal testing is a nice tool in the paper, but you have also to keep in mind to add materials' characterization and other recommended tests.
We appreciate your comments that help us to improve our paper. We have considered all your recommendations and wishes above and have added some information about mechanical properties, obtaining methods, etc.
Please see the attachment with corrected paper.

Reviewer 2 Report
Amazing work. I am in favour of accepting the present paper, although I would suggest a number of different tests that if you are able to run, it could improve the paper a lot.
The assays I would suggest carrying out in the different materials are:
1. Mechanical testing.
2. Alamar Blue or life-dead assays.
3. Degradation assay in enzymatic and non-enzymatic conditions.
I would be very happy to see some of these assays in the paper.
Author Response
Dear Sir/Madam,
We have received your review comments and carefully familiarized ourselves with them. We appreciate you reading our paper and providing us with helpful comments. Your comments will help us improve our paper.
Below are our responses to each of your comments:
- Mechanical testing.
Measurements of the mechanical properties (breaking stress, Young's modulus, and strain at break) of the fibers were carried out using an Instron 5943 tensile testing machine. The fiber length used as the base was 100 mm, and the fiber loading rate was maintained at 10 mm/min. Before testing, it was ensured that the fibers had been stored in normal climatic conditions for a minimum of 24 hours, with a relative humidity of 66% and a temperature of 20 ± 2°C (lines 161-166). The results are observed in Table 1 (line 272).
- Alamar Blue or life-dead assays.
Alamar Blue or life-dead assays were conducted.
Sterile samples of PLA conduits and chitosan-based fibers were placed into the wells, where the FetMSCs cell suspension was added (Fig.1). After two days (due to the time limit, the experiment was carried out quickly), the nutrient medium was removed, and the cells were washed once with a PBS solution. The cell viability in the presence of samples was assessed using the LIVE/DEAD Viability/Cytotoxicity Kit (Invitrogen, USA). The principle of the method is to stain the cells with two fluorescent dyes, namely Calcein AM and Ethidium homodimer. Calcein AM was used to mark the cytoplasm of living cells, while Ethidium homodimer was used to mark the nuclei of cells that have passed away and damaged membranes.
The findings were observed through a confocal microscope (Olympus FV3000, Japan, and Leica SP8, Germany). The living cells exhibited green fluorescence due to Calcein AM, while the nuclei of dead cells displayed red fluorescence due to Ethidium homodimer. As shown in Fig.2 all cells were stained green in the presence of the examined samples.
Fig. 1. Samples of conduits and fibers in the well.
(a) |
(b) |
Fig. 2. Cells in a well with (a) a PLA conduit and (b) chitosan fibers.
- Degradation assay in enzymatic and non-enzymatic conditions.
We have conducted extensive degradation analysis of chitosan composite fibers with chitin nanofibrils and polylactide conduits in PBS and PBS with lysozyme. We have obtained interesting voluminous results and devoted a separate article. Unfortunately, the article is currently under review in the journal, so I am not allowed to include degradation data.
Please see the attachment with corrected paper.

Reviewer 3 Report
In this study, Nurjemal A. Tagandurdyyeva et al developed a nerve conduit consisting of a combination of poly(L-lac-22 tide) nanofibers and chitosan composite fibers with chitin nanofibrils. The conduit design imitated the oriented architecture of the nerve, facilitated electrical communication between the damaged nerve's ends, and promoted the direction of nerve growth, showing promising for peripheral nerve regeneration. The topic as such is suitable for the journal, but some issues should be further addressed
1. How to fix the chitosan fiber in the PLA conduit? What is the stability of the chitosan fiber?
2. Why did you choose a material for polylactic acid nanofibers as a conduit substrate? Why not consider other natural or synthetic materials?
3. The 2 of CO2 in lines 142-148 should be subscripted.
4. Why ethanol was used in line 150?
5. What is the basis for the diameter of the fibers filled in the conduit?
6. There is no scale bar or any dimensional reference object in Figure 4. The conduit in Figure 4e seems to be longer than the other groups, and it is better to uniformize the size of the picture.
7. How often can Human mesenchymal stromal cells crawl an entire conduit along the fiber? Is there a significant time difference between hollow conduit and nanofibrils-filled conduit?
8. What is the conductivity of chitosan fibers?
9. Has the conduit degraded in the body after 16 weeks? How long does its degradation cycle take?
Minor revision needs to be done
Author Response
Dear Sir/Madam,
We have received your review comments and carefully familiarized ourselves with them. We appreciate you reading our paper and providing us with helpful comments. Your comments will help us improve our paper.
Below are our responses to each of your comments:
- How to fix the chitosan fiber in the PLA conduit? What is the stability of the chitosan fiber?
Before the surgical procedure, the chitosan fibers were incorporated into the PLA conduits. The fibers were then fastened to the conduits' inner surface through surgical sutures located at the distal and proximal ends. Remarkably, the materials' excellent wettability, which is attributed to their chemical properties and physical structure, facilitated the firm adhesion of the chitosan fibers to the conduits' surface throughout their entire length. Notably, this assertion has been corroborated by in vitro experiments.
- Why did you choose a material for polylactic acid nanofibers as a conduit substrate? Why not consider other natural or synthetic materials?
Polylactide was chosen due to its biocompatibility and bioresorbability and satisfactory mechanical properties. While other polymers such as polyglycolide and polycaprolactone tend to lose their mechanical properties quickly, it is crucial for us to preserve the mechanical properties of the conduit for a more extended period to prevent compression of the newly formed axons.
Although natural polymers offer significant advantages, creating conduits based on them can be challenging due to the requirement of achieving adequate mechanical properties. Hence, we incorporated chitosan fibers, a natural material with a neuroprotective effect, into the synthetic material - polyglycolide, which has higher mechanical properties. This approach allowed us to create a conducive microenvironment and structure for nerve regeneration.
- The 2 of CO2in lines 142-148 should be subscripted.
Thank you for bringing the note to my attention. All text has been corrected.
- Why was ethanol used in line 150?
The scanning electron microscope (SEM) is a highly regarded research instrument that utilizes a focused electron beam to generate images of a sample. To ensure accurate results, SEM samples must be completely dry. As a result, biological samples such as cells, tissues, and organisms must undergo chemical fixation to preserve and stabilize their structure. It is recommended that ethanol be used for dehydration purposes [Fischer, E.R.; Hansen, B.T.; Nair, V.; Hoyt, F.H.; Dorward, D.W. Scanning Electron Microscopy. Curr Protoc Microbiol 2012, 25, doi:10.1002/9780471729259.mc02b02s25; Shehadat, S. Al; Gorduysus, M.O.; Hamid, S.S.A.; Abdullah, N.A.; Samsudin, A.R.; Ahmad, A. Optimization of Scanning Electron Microscope Technique for Amniotic Membrane Investigation: A Preliminary Study. Eur J Dent 2018, 12, 574–578, doi:10.4103/ejd.ejd_401_17].
- What is the basis for the diameter of the fibers filled in the conduit?
The selection of the fiber diameter was made with consideration for the optimum and efficient amount of chitosan required to cover the conduit surface. Furthermore, the fibers were chosen for their ease of use by surgeons during surgical procedures. Presently, there is ongoing experimentation with monofilament fibers measuring up to 25 µm in diameter, aimed at evaluating their impact on axonal regeneration.
- There is no scale bar or any dimensional reference object in Figure 4. The conduit in Figure 4e seems to be longer than the other groups, and it is better to uniformize the size of the picture.
Thank you very much for bringing this to my attention. I have adjusted to ensure that the picture sizes are now consistent throughout.
- How often can Human mesenchymal stromal cells crawl an entire conduit along the fiber? Is there a significant time difference between hollow conduit and nanofibrils-filled conduit?
The existence of appropriate pores in conduits greatly facilitates the diffusion of essential nutrients, such as oxygen, blood vessels, and growth factors, as well as the removal of debris and waste material from the conduit. This, in turn, leads to a faster process of nerve growth and regeneration [Jahromi, M.; Razavi, S.; Bakhtiari, A. The Advances in Nerve Tissue Engineering: From Fabrication of Nerve Conduit to in Vivo Nerve Regeneration Assays. J Tissue Eng Regen Med 2019, 13, 2077–2100, doi:10.1002/term.2945; Sarker, M.; Naghieh, S.; McInnes, A.D.; Schreyer, D.J.; Chen, X. Strategic Design and Fabrication of Nerve Guidance Conduits for Peripheral Nerve Regeneration. Biotechnol J 2018, 13, e1700635, doi:10.1002/biot.201700635]. The size of pores in conduits is a crucial structural parameter that affects the proper regeneration of nerves. Specifically, the pore size of a conduit is a significant factor that determines the nerve route and promotes nerve growth [Oh, S.H.; Kim, J.R.; Kwon, G.B.; Namgung, U.; Song, K.S.; Lee, J.H. Effect of Surface Pore Structure of Nerve Guide Conduit on Peripheral Nerve Regeneration. Tissue Eng Part C Methods 2013, 19, 233–243, doi:10.1089/ten.TEC.2012.0221].
- What is the conductivity of chitosan fibers?
After being exposed to a biological environment, the conductivity of a fiber consisting of chitosan has been measured to be 3*10-6 S/m at a voltage of 100V. However, in its dry state, the fiber's conductivity is approximately 10-9 S/m.
The article provides a comprehensive description of the conductivity of chitosan films containing chitin nanofibrils studied by coauthors [Dresvyanina, E.N.; Kodolova-Chukhontseva, V. V; Bystrov, S.G.; Dobrovolskaya, I.P.; Vaganov, G. V; Smirnova, N. V; Kolbe, K.A.; Kamalov, A.M.; Ivan’kova, E.M.; Morganti, P.; et al. Influence of Surface Morphology of Chitosan Films Modified by Chitin Nanofibrils on Their Biological Properties. Carbohydr Polym 2021, 262, 117917, doi:10.1016/j.carbpol.2021.117917].
- Has the conduit degraded in the body after 16 weeks? How long does its degradation cycle take?
The duration of the conduit strength retention is documented to be three months, while its complete degradation can take anywhere from 18 to 24 months.
When chitosan fibers were introduced directly into the endomysium, resorption was observed within 30 days. However, when implanted into perimysium after 45 days of exposure no signs of fiber destruction were registered. Notably, the introduction of chitin nanofibrils has been shown to increase the time of fiber resorption due to the longer resorption of nanofibrils.
Extensive degradation analysis of chitosan composite fibers with chitin nanofibrils in PBS and PBS with lysozyme was conducted. The results showed increasing resorption duration with an increasing amount of chitin nanofibrils in the fibers.
We have obtained interesting voluminous results and devoted a separate article. Unfortunately, the article is currently under review in the journal, so I am not allowed to include degradation data.
Round 2
Reviewer 1 Report
Dear authors,
After revision the paper has been improved. Pay attention to comma when you have numbers with decimals in English: dot instead of comma. In some tables you used commas, in other tables dots. I am ready to accept the paper.
The English requires minor revision as stated in the comments (dot instead of comma for decimal numbers).